# Trends in prevalence of acute stroke impairments: A population-based cohort study using the South London Stroke Register

**Amanda Clery** [1] *, **Ajay Bhalla** [1,2], **Anthony G. Rudd** [1,2], **Charles D. A. Wolfe** [1,3,4], **Yanzhong Wang** [1,3,4]

**1** School of Population Health and Environmental Sciences, King's College London, London, United Kingdom, **2** Guy's and St Thomas' NHS Foundation Trust, London, United Kingdom, **3** National Institute for Health Research (NIHR) Biomedical Research Centre (BRC), Guy's and St Thomas' NHS Foundation Trust and King's College London, London, United Kingdom, **4** NIHR Applied Research Collaboration (ARC) South London, London, United Kingdom

* amanda.clery@kcl.ac.uk

**Data Availability Statement:** The raw data for this study contain both personally identifiable and confidential clinical data. The participants of the study did not consent to sharing the information

## Abstract

### Background

Acute stroke impairments often result in poor long-term outcome for stroke survivors. The aim of this study was to estimate the trends over time in the prevalence of these acute stroke impairments.

### Methods and findings

All first-ever stroke patients recorded in the South London Stroke Register (SLSR) between 2001 and 2018 were included in this cohort study. Multivariable Poisson regression models with robust error variance were used to estimate the adjusted prevalence of 8 acute impairments, across six 3-year time cohorts. Prevalence ratios comparing impairments over time were also calculated, stratified by age, sex, ethnicity, and aetiological classification (Trial of Org 10172 in Acute Stroke Treatment [TOAST]). A total of 4,683 patients had a stroke between 2001 and 2018. Mean age was 68.9 years, 48% were female, and 64% were White. After adjustment for demographic factors, pre-stroke risk factors, and stroke subtype, the prevalence of 3 out of the 8 acute impairments declined during the 18-year period, including limb motor deficit (from 77% [95% CI 74%–81%] to 62% [56%–68%], $p < 0.001$), dysphagia (37% [33%–41%] to 15% [12%–20%], $p < 0.001$), and urinary incontinence (43% [39%–47%) to 29% [24%–35%], $p < 0.001$). Declines in limb impairment over time were 2 times greater in men than women (prevalence ratio 0.73 [95% CI 0.64–0.84] and 0.87 [95% CI 0.77–0.98], respectively). Declines also tended to be greater in younger patients. Stratified by TOAST classification, the prevalence of all impairments was high for large artery atherosclerosis (LAA), cardioembolism (CE), and stroke of undetermined aetiology. Conversely, small vessel occlusions (SVOs) had low levels of all impairments except for limb motor impairment and dysarthria. While we have assessed 8 key acute stroke impairments, this study is limited by a focus on physical impairments, although cognitive

publicly, and our ethical approvals require strict information governance procedures. Requests for data access for academic use should be made to the South London Stroke Register, where data will be made available subject to academic review and acceptance of a data-sharing agreement. Information can be found/requested through the following link: https://www.kcl.ac.uk/lsm/research/divisions/hscr/research/groups/stroke.

**Funding:** Authors CW and YW received funding from the National Institute for Health Research (NIHR) Biomedical Research Centre (BRC), Guy's and St Thomas' NHS Foundation Trust and King's College London, London, United Kingdom (www.guysandstthomasbrc.nihr.ac.uk) and the NIHR Applied Research Collaboration (ARC) South London, London, United Kingdom (www.arc-sl.nihr.ac.uk). The funders had no role in study design, data collection and analysis, decision to publish, or preparation of the manuscript.

**Competing interests:** I have read the journal's policy and the authors of this manuscript have the following competing interests: YW is a member of the Editorial Board of *PLOS Medicine*, but had no role in the peer review of this paper.

**Abbreviations:** AF, atrial fibrillation; CE, cardioembolism; LAA, large artery atherosclerosis; MI, myocardial infarction; NIHSS, National Institutes of Health Stroke Scale; PICH, primary intracerebral haemorrhage; SAH, subarachnoid haemorrhage; SLSR, South London Stroke Register; SSNAP, Sentinel Stroke National Audit Programme; STROBE, Strengthening The Reporting of OBservational studies in Epidemiology; SVO, small vessel occlusion; TIA, transient ischaemic attack; TOAST, Trial of Org 10172 in Acute Stroke Treatment.

impairments are equally important to understand. In addition, this is an inner-city cohort, which has unique characteristics compared to other populations.

## Conclusions

In this study, we found that stroke patients in the SLSR had a complexity of acute impairments, of which limb motor deficit, dysphagia, and incontinence have declined between 2001 and 2018. These reductions have not been uniform across all patient groups, with women and the older population, in particular, seeing fewer reductions.

## Author summary

### Why was this study done?

- Stroke is one of the top 5 causes of disability globally.
- We do not know how the different types of disability caused by stroke have improved or changed over time.

### What did the researchers do and find?

- We analysed the changes in the prevalence of 8 different stroke impairments between 2001 and 2018 in a total of 4,683 stroke patients.
- Over time, fewer people experienced limb impairment, dysphagia (swallowing difficulties), and incontinence, but the other 5 impairments did not decline over time. These were visual field defect, neglect, sensory loss, dysphasia, and dysarthria.
- The people who tended to continue experiencing these impairments over time were older and female patients.

### What do these findings mean?

- The type and number of disabilities that stroke patients in our study population face has changed over time.
- This has implications for how patients are cared for by clinicians in the short term and how the needs of stroke survivors are addressed by public health policy in the long term.

## Introduction

Stroke incidence has declined by 30% in the United Kingdom over the past 2 decades [1]. However, stroke remains responsible for causing life-changing impairments in its survivors, leading to it being one of the top 5 causes of disability-adjusted life-years in high-income countries and globally in 2017 [2]. Impairments after acute stroke are frequent and usually multiple, with studies reporting up to 80% prevalence of limb impairments and between 20% and 60%

for speech, language, and visual impairments [3–7]. Research has tended to focus on the prevalence and incidence of impairments in data collected over short timeframes, and research exploring these impairments over time is lacking.

While impairments improve for many patients after stroke and some see full recovery [8–10], many patients are left with life-long disability, affecting quality of life. Acute stroke impairments reflecting stroke severity are also the strongest predictors for this long-term disability [11–14].

Given the high prevalence and complexity of impairments in the acute phase of stroke, and their implications for the future life of people living with stroke, an understanding of the burden of these impairments in the long term is crucial not only for patients and clinicians but also policymakers in order to shape future care and service provision.

Despite studies into the prevalence of a variety of acute stroke impairments, there has been a lack of data on long-term trends in prevalence, limiting our understanding of any changes in the patterns of these impairments over time. Characteristics of the stroke population and preventive medicine have changed over time, which contribute to the severity of stroke [15–17].

In this study, we used a large, population-based cohort from London, UK, to assess longitudinal trends in the prevalence of acute stroke impairments between 2001 and 2018, as well as the prevalence in population subgroups. Additionally, we explored the associations between acute stroke impairments and aetiological subtype of stroke over the duration of the study.

## Methods

### Study population

This study used data from the South London Stroke Register (SLSR), an ongoing, prospective, population-based register that has recorded all first-ever strokes within Lambeth and Southwark, inner-city South London, UK, since 1 January 1995. At the 2011 UK Census, the source population was 357,308, with 56% White, 25% Black (14% Black African, 7% Black Caribbean, and 4% Other Black), and 18% Other ethnic groups. Details on the notification of patients have been described elsewhere [18]. Analysis for this study was planned in July 2019.

### Data collection

All first-ever stroke patients registered between 1 January 2001 and 31 December 2018 were included in this analysis. All follow-up data—collected prospectively by trained nurses, doctors, and fieldworkers—included in this study were collected by 31 January 2019.

Data collected at initial assessment included the following:

1. Demographic factors: age at time of stroke; sex; ethnicity self-defined as per census question and categorized into White, Black (including African, Caribbean, and Other Black ethnic groups), and Other (including Asian, Pakistani, Indian, Bangladeshi, Chinese, and any other ethnic groups); and employment (Registrar General's occupational codes, grouped into manual or nonmanual)

2. Pre-stroke risk factors: hypertension (systolic blood pressure >140 mmHg or diastolic >90 mmHg), myocardial infarction (MI), atrial fibrillation (AF), diabetes, high cholesterol (total cholesterol concentration ≥6 mmol/L or ≥232 mg/dL), vascular disease (including previous transient ischaemic attack [TIA], ischaemic heart disease, or peripheral vascular disease), and smoking status (current, ex-, or non-smoker)

3. Pre-stroke prescriptions for primary stroke prevention: antihypertensives for those with hypertension, anticoagulation for those with AF, statins for those with high cholesterol or vascular disease, and antiplatelets for those with vascular disease

4. Acute-phase stroke characteristics: stroke subtype as per the Trial of Org 10172 in Acute Stroke Treatment (TOAST) classification categorized into LAA, cardioembolism (CE), small vessel occlusion (SVO), other aetiologies (such as rare vasculopathies or haematologic disorders), and undetermined aetiologies (despite extensive evaluation, also includes those with multiple potential causes) [19]; haemorrhagic subtypes of primary intracerebral haemorrhage (PICH) and subarachnoid haemorrhage (SAH); overall stroke severity measured using the National Institutes of Health Stroke Scale (NIHSS) scored from 0 (mild) to 42 (severe) [20]; visual field defect (NIHSS item 3); neglect (inattention, item 11); upper- and lower-limb motor deficits (items 5 and 6 combined); sensory loss (item 8); dysphasia (item 9); and dysarthria (item 10). Individual NIHSS items were only collected in the SLSR from 2004, prior to which impairments were collected using comparable methods. Additional impairments collected were dysphagia assessed formally with the swallow test [21] and incontinence. All acute-phase stroke characteristics were collected by fieldworkers from clinical notes recorded by the admitting clinical team.

## Statistical analysis

Patients were categorised based on year of stroke into six 3-year cohorts: 2001–2003, 2004–2006, 2007–2009, 2010–2012, 2013–2015, and 2016–2018.

All variables from the initial data collection were summarised, stratified by year of stroke, and analysed using the chi-squared test for trend for categorical variables and ANOVA for continuous variables.

The prevalence of each acute impairment was summarised using percentages for each time cohort, and 95% confidence intervals were calculated for each, as well as the median total number of impairments in each timeframe. Using Poisson regression models with robust error variance in a complete case analysis, these prevalence estimates were adjusted for age, sex, ethnicity, TOAST classification, and pre-stroke risk factors (hypertension, MI, AF, TIA, diabetes, high cholesterol, and smoking status).

Prevalence ratios and their 95% confidence intervals over time were also calculated for each impairment between the oldest cohort (2001–2003) and the most recent (2016–2018).

The trends in the prevalence of impairments over time were also analysed by subgroup for age, sex, ethnicity, and TOAST subtype. Analysis by age, sex, and ethnicity was adjusted for TOAST subtype and all pre-stroke risk factors (hypertension, MI, AF, TIA, diabetes, high cholesterol, and smoking status). Analysis by TOAST subtype was adjusted for age, sex, and ethnicity only because of lack of power. Similar methods as given earlier were used to calculate adjusted prevalence and prevalence ratios along with the 95% confidence intervals.

Crude associations between TOAST classification and prevalence of acute impairments were also assessed and tested using the chi-squared test for trend.

Except for the summary statistics, all subsequent analyses using TOAST subtype excluded the "other aetiologies" category, as this was a very small subgroup and any analysis lacked power.

## Ethics

All patients and/or relatives gave written informed consent to participate in the study. Ethical approval for the study was obtained from the ethics committees of Guy's and St Thomas' NHS Foundation Trust, King's College Hospital Foundation Trust, Queen's Square, St George's University Hospital, and Westminster Hospital (No. EC01 020).

## Results

A total of 4,683 people had a first-ever stroke between 2001 and 2018. Table 1 shows that from the earliest cohort (2001–2003) until the latest cohort (2016–2018), first-ever stroke patients became slightly younger (mean age 69.6 to 68.5 years, Table 1). Mean age was higher in White patients (71.7 years) compared to Black patients (64.0 years). There was an increase in the number of Black patients in the study population (from 21% to 40%), and a shift from majority skilled manual labour (67%) to majority (54%) nonmanual labour was seen.

**Table 1. Demographic, clinical, and process-of-care factors for first-ever stroke patients in the SLSR 2001–2018, stratified by year of stroke.**

| Factor | All years (N = 4,683) | Year of stroke | | | | | | p-Value[1] |
|---|---|---|---|---|---|---|---|---|
| | | 2001–2003 (N = 828) | 2004–2006 (N = 1,070) | 2007–2009 (N = 836) | 2010–2012 (N = 674) | 2013–2015 (N = 593) | 2016–2018 (N = 682) | |
| **Mean age (SD)** | 68.9 (15.7) | 69.6 (15.0) | 69.0 (15.3) | 70.0 (15.7) | 68.2 (16.1) | 67.5 (16.2) | 68.5 (16.2) | 0.037 |
| **Female** | 2,240 (47.8) | 415 (50.1) | 498 (46.5) | 420 (50.2) | 318 (47.2) | 272 (45.9) | 317 (46.5) | 0.167 |
| **Ethnicity** | | | | | | | | <0.001 |
| White | 2,903 (63.7) | 577 (73.5) | 697 (66.9) | 565 (69.1) | 375 (56.6) | 330 (57.6) | 359 (53.3) | |
| Black | 1,335 (29.3) | 164 (20.9) | 266 (25.5) | 203 (24.8) | 238 (35.9) | 195 (34.0) | 269 (39.9) | |
| Other | 317 (7.0) | 44 (5.6) | 79 (7.6) | 50 (6.1) | 50 (7.5) | 48 (8.4) | 46 (6.8) | |
| **Employment: Skilled manual labour** | 1,824 (59.6) | 473 (67.3) | 561 (64.6) | 336 (56.7) | 225 (53.1) | 150 (50.3) | 79 (45.7) | <0.001 |
| **Hypertension** | 2,984 (65.4) | 487 (61.3) | 695 (65.8) | 524 (64.1) | 437 (66.4) | 387 (67.0) | 454 (68.5) | 0.007 |
| **MI** | 464 (10.3) | 95 (11.9) | 90 (8.5) | 68 (8.4) | 60 (9.3) | 54 (9.8) | 97 (14.8) | 0.042 |
| **AF** | 755 (16.7) | 121 (15.3) | 141 (13.4) | 120 (14.7) | 117 (18.2) | 131 (23.0) | 125 (19.4) | <0.001 |
| **Diabetes** | 1,052 (23.1) | 143 (18.2) | 226 (21.6) | 175 (21.2) | 158 (23.8) | 143 (25.0) | 207 (31.0) | <0.001 |
| **High cholesterol** | 1,389 (30.7) | 119 (15.4) | 259 (24.6) | 244 (30.1) | 233 (35.8) | 252 (44.3) | 282 (42.6) | <0.001 |
| **Vascular disease** | 932 (20.4) | 247 (30.9) | 286 (27.1) | 162 (19.8) | 75 (11.5) | 62 (10.8) | 100 (15.0) | <0.001 |
| **Smoking status** | | | | | | | | <0.001 |
| Never | 1,675 (40.5) | 274 (38.0) | 358 (36.9) | 278 (37.9) | 279 (45.6) | 243 (46.4) | 243 (42.3) | |
| Ex | 1,296 (31.3) | 211 (29.3) | 317 (32.6) | 234 (31.9) | 173 (28.3) | 164 (31.3) | 197 (34.3) | |
| Current | 1,166 (28.2) | 236 (32.7) | 296 (30.5) | 222 (30.2) | 160 (26.1) | 117 (22.3) | 135 (23.5) | |
| **Antihypertensives** | 1,751 (59.9) | 331 (69.5) | 507 (73.8) | 304 (58.8) | 200 (47.6) | 183 (49.1) | 226 (50.3) | <0.001 |
| **Anticoagulation** | 188 (25.5) | 22 (18.5) | 27 (19.1) | 26 (22.4) | 30 (26.8) | 34 (26.8) | 49 (39.8) | <0.001 |
| **Statins** | 1,129 (60.7) | 95 (33.0) | 227 (56.5) | 221 (67.0) | 184 (70.2) | 174 (65.4) | 228 (72.8) | <0.001 |
| **Antiplatelets** | 571 (62.0) | 151 (61.9) | 201 (70.5) | 100 (62.1) | 40 (55.6) | 33 (54.1) | 46 (46.9) | <0.001 |
| **Stroke subtype: Ischemic** | 3,819 (82.2) | 647 (79.7) | 864 (81.9) | 705 (84.6) | 571 (85.0) | 484 (81.8) | 548 (80.5) | 0.617 |
| **Stroke subtype: TOAST classification** | | | | | | | | 0.610 |
| LAA | 376 (9.1) | 49 (6.2) | 98 (9.6) | 97 (12.6) | 56 (11.0) | 33 (6.5) | 43 (8.5) | |
| CE | 882 (21.5) | 174 (21.9) | 211 (20.6) | 142 (18.4) | 105 (20.6) | 132 (25.8) | 118 (23.4) | |
| SVO | 842 (20.5) | 178 (22.4) | 186 (18.2) | 158 (20.5) | 136 (26.7) | 85 (16.6) | 99 (19.6) | |
| Other | 84 (2.0) | 17 (2.1) | 30 (2.9) | 13 (1.7) | 6 (1.2) | 9 (1.8) | 9 (1.8) | |
| Undefined | 1,172 (28.5) | 212 (26.7) | 309 (30.2) | 243 (31.6) | 118 (23.2) | 154 (30.1) | 136 (27.0) | |
| PICH | 550 (13.4) | 107 (13.5) | 139 (13.6) | 92 (11.9) | 67 (13.2) | 71 (13.9) | 74 (14.7) | |
| SAH | 205 (5.0) | 58 (7.3) | 49 (4.8) | 25 (3.2) | 21 (4.1) | 27 (5.3) | 25 (5.0) | |
| **NIHSS score, median (IQR)** | 6.0 (3.0–13.0) | 7.0 (3.0–12.0) | 6.0 (3.0–14.0) | 7.0 (4.0–16.0) | 5.0 (2.0–10.0) | 5.0 (3.0–11.0) | 5.0 (2.0–10.0) | <0.001 |

[1]Chi-squared test for trend for categorical variables, ANOVA for normal continuous variables, and Kruskal-Wallis for non-normal continuous variables.

**Abbreviations:** AF, atrial fibrillation; CE, cardioembolism; LAA, large artery atherosclerosis; MI, myocardial infarction; NIHSS, National Institutes of Health Stroke Scale; PICH, primary intracerebral haemorrhage; SAH, subarachnoid haemorrhage; SLSR, South London Stroke Register; SVO, small vessel occlusion; TOAST, Trial of Org 10172 in Acute Stroke Treatment

The prevalence of all pre-stroke risk factors increased in the study population over time, except for vascular disease and current smokers, which we show to have decreased. The greatest increase was seen in high cholesterol from 15% in 2001–2003 to 43% in 2016–2018 ($p <$ 0.001). However, pre-stroke medications for these risk factors have not all shown similar increases. Of the patients who have hypertension, there has been a decrease in those taking antihypertensives from 70% to 50% ($p < 0.001$). There has also been a decline in the use of antiplatelets from 62% to 47% over time ($p < 0.001$). However, we have shown a doubling in the use of anticoagulation for AF patients from 19% to 40%. There has also been an increase in the use of statins from 33% to 73% over the 18-year study period (Table 1).

There were no changes in the prevalence of each aetiological subtype between 2001 and 2008, as per the TOAST classification ($p = 0.61$). There has been a decrease in stroke severity from a median NIHSS score of 7 (IQR 3–12) to a score of 5 (IQR 2–10, $p < 0.001$) across all stroke patients.

All characteristics had <5% missing data except for smoking status (11.7%), stroke subtype (12.2%), and total NIHSS score (14.3%). Acute impairment variables had missing values between 9.3% and 11.7% except dysphagia, which had 17.0%, and incontinence with only 5.7%.

Using a total of 3,381 complete cases, after adjustment for age, sex, ethnicity, pre-stroke risk factors, and aetiological stroke subtype, the prevalence of 3 out of the 8 acute impairments studied declined over time: limb motor deficit (from 77.4% to 61.7%), dysphagia (37.3% to 15.4%), and incontinence (42.7% to 29.2%), summarised in Fig 1. Visual field defect, neglect, sensory loss, dysphasia, and dysarthria saw no declines in prevalence over time. Consistently, the most common impairment over time was limb motor deficit, which remained the most prevalent in the most recent cohort, 2016–2018 (61.7%; 95% CI 56.3%–67.6%; Table 2). The next most common impairment was dysarthria at 45.2% (95% CI 38.7%–52.7%). The least prevalent impairment was dysphagia (15.4%; 95% CI 11.8%–20.1%). In 2016–2018, the median number of impairments for a patient was 2 out of the 8 studied (Table 2). The greatest decline from 2001–2003 to 2016–2018 was seen in dysphagia, which more than halved from 37% prevalence to 15% (adjusted prevalence ratio: 0.41; 95% CI 0.32–0.54; Fig 1).

After stratifying by sex and adjustment for other demographic and pre-stroke risk factors and stroke subtype, the decline over time in limb impairments was 2 times greater for males than females (Fig 2); 76% of male patients who had a stroke in 2001–2003 had upper limb impairment, but by 2016–2018, there was a 20% relative decline (prevalence ratio 0.73; 95% CI 0.64–0.84). Comparatively, female patients started with a slightly higher prevalence of 79% in 2001–2003, which was followed by only a 10% relative decline to 69% by 2016–2018 (Fig 2). In all other impairments, there was also a tendency for men to show greater declines than women, but their confidence intervals overlapped.

Fig 3 shows a similar trend stratified by age; patients younger than 65 years old tended to show greater declines than those 65 years or older, although all confidence intervals overlapped.

After stratifying by ethnicity, we showed that, while declines were all similar between Black and White patients, the declines tended to be greater in those who were Black—although, as with age, all confidence intervals overlapped (Fig 4).

Finally, with stratification by TOAST classification, we show that the decline in incontinence was driven by LAA and undefined aetiologies (prevalence ratio: 0.42 [95% CI 0.23–0.79] and 0.64 [0.49–0.84], respectively). For limb motor deficit and dysphagia, the declines were similar in all subgroups (Fig 5). We also looked at the association between the prevalence of each impairment and stroke subtype (Table 3). For all but SAH, limb motor deficit remained the most prevalent impairment ranging from 74.5% in undetermined aetiology to 83.3% for

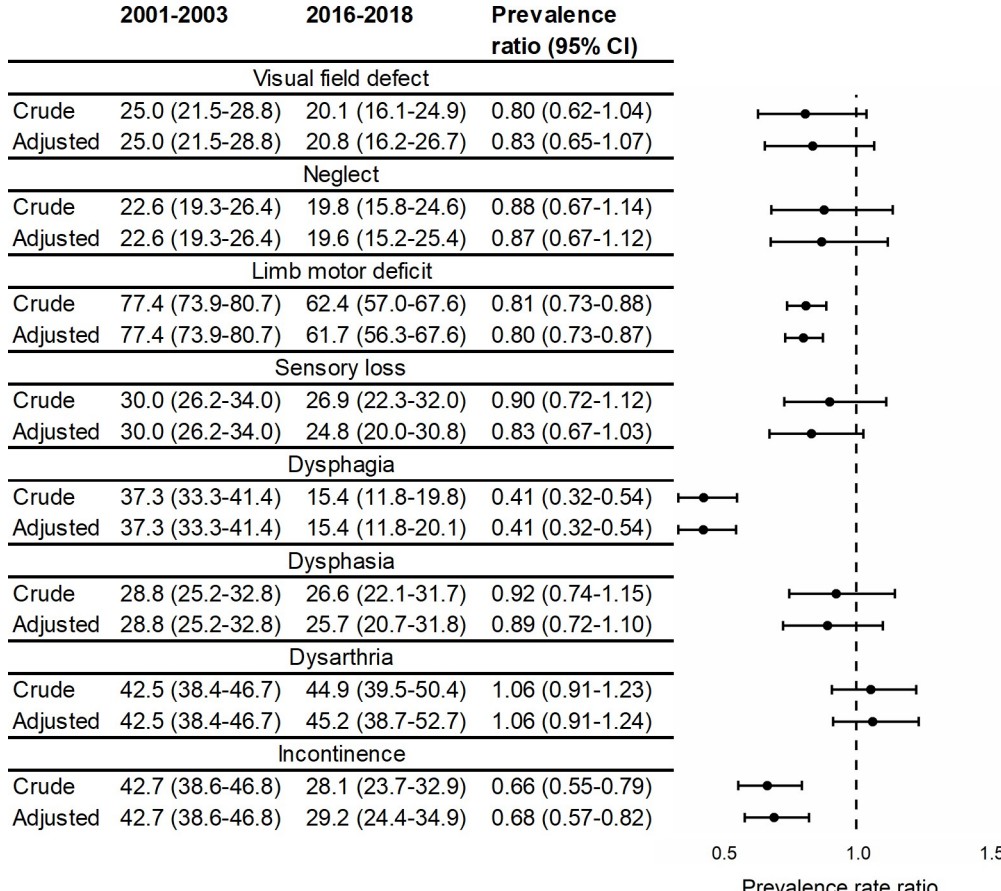

| | 2001-2003 | 2016-2018 | Prevalence ratio (95% CI) |
|---|---|---|---|
| **Visual field defect** | | | |
| Crude | 25.0 (21.5-28.8) | 20.1 (16.1-24.9) | 0.80 (0.62-1.04) |
| Adjusted | 25.0 (21.5-28.8) | 20.8 (16.2-26.7) | 0.83 (0.65-1.07) |
| **Neglect** | | | |
| Crude | 22.6 (19.3-26.4) | 19.8 (15.8-24.6) | 0.88 (0.67-1.14) |
| Adjusted | 22.6 (19.3-26.4) | 19.6 (15.2-25.4) | 0.87 (0.67-1.12) |
| **Limb motor deficit** | | | |
| Crude | 77.4 (73.9-80.7) | 62.4 (57.0-67.6) | 0.81 (0.73-0.88) |
| Adjusted | 77.4 (73.9-80.7) | 61.7 (56.3-67.6) | 0.80 (0.73-0.87) |
| **Sensory loss** | | | |
| Crude | 30.0 (26.2-34.0) | 26.9 (22.3-32.0) | 0.90 (0.72-1.12) |
| Adjusted | 30.0 (26.2-34.0) | 24.8 (20.0-30.8) | 0.83 (0.67-1.03) |
| **Dysphagia** | | | |
| Crude | 37.3 (33.3-41.4) | 15.4 (11.8-19.8) | 0.41 (0.32-0.54) |
| Adjusted | 37.3 (33.3-41.4) | 15.4 (11.8-20.1) | 0.41 (0.32-0.54) |
| **Dysphasia** | | | |
| Crude | 28.8 (25.2-32.8) | 26.6 (22.1-31.7) | 0.92 (0.74-1.15) |
| Adjusted | 28.8 (25.2-32.8) | 25.7 (20.7-31.8) | 0.89 (0.72-1.10) |
| **Dysarthria** | | | |
| Crude | 42.5 (38.4-46.7) | 44.9 (39.5-50.4) | 1.06 (0.91-1.23) |
| Adjusted | 42.5 (38.4-46.7) | 45.2 (38.7-52.7) | 1.06 (0.91-1.24) |
| **Incontinence** | | | |
| Crude | 42.7 (38.6-46.8) | 28.1 (23.7-32.9) | 0.66 (0.55-0.79) |
| Adjusted | 42.7 (38.6-46.8) | 29.2 (24.4-34.9) | 0.68 (0.57-0.82) |

**Fig 1. Prevalence ratios for acute stroke impairments over time, crude and adjusted for age, sex, ethnicity, TOAST subtype, and pre-stroke risk factors (hypertension, MI, AF, TIA, diabetes, high cholesterol, and smoking status) (N = 3,381).** AF, atrial fibrillation; MI, myocardial infarction; TIA, transient ischaemic attack; TOAST, Trial of Org 10172 in Acute Stroke Treatment.

PICH. The most prevalent impairment in SAHs was incontinence (51.8%). SVOs had relatively low levels of all impairments except for limb motor deficit, with a high prevalence of 79%, and dysarthria. Conversely, strokes of undetermined aetiologies had high levels of all impairments, comparable to LAA and CEs.

## Discussion

This study used population-based data to describe the trends in the prevalence of acute stroke impairments over time. We have shown that stroke patients have high levels of acute impairments, particularly those relating to limb function. Limb motor deficit, incontinence, and dysphagia have all declined over the past 18 years. We also show that this decline is greater in men, and the younger stroke population.

The declines in limb motor deficit, incontinence, and dysphagia are present even after adjustment for all demographic factors, pre-stroke risk factors, and stroke subtype. As has been previously established, these impairments are essential indicators of long-term disability and quality of life [14]. Therefore, a decline in these impairments is promising to see future improvements in the lives of stroke survivors. However, we have also shown that the prevalence of acute impairments remained high, despite these declines. This has important

**Table 2. The trends in the prevalence of acute impairments over time, adjusted for age, sex, ethnicity, TOAST classification, and pre-stroke risk factors (hypertension, MI, AF, TIA, diabetes, high cholesterol, and smoking status) (*N* = 3,381).**

| Acute impairment | Prevalence, % (95% CI) | | | | | | *p*-Value[1] |
|---|---|---|---|---|---|---|---|
| | 2001–2003 (*N* = 633) | 2004–2006 (*N* = 905) | 2007–2009 (*N* = 631) | 2010–2012 (*N* = 420) | 2013–2015 (*N* = 398) | 2016–2018 (*N* = 394) | |
| Visual field defect | 25.0 (21.5–28.8) | 20.7 (17.2–24.8) | 23.5 (19.5–28.4) | 24.2 (19.3–30.4) | 27.0 (21.5–33.8) | 20.8 (16.2–26.7) | 0.848 |
| Neglect | 22.6 (19.3–26.4) | 22.8 (18.9–27.4) | 30.1 (25.0–36.3) | 21.0 (16.5–26.8) | 19.3 (15.0–24.9) | 19.6 (15.2–25.4) | 0.109 |
| Limb motor deficit | 77.4 (73.9–80.7) | 78.8 (74.7–83.1) | 77.7 (73.3–82.4) | 70.4 (65.3–75.9) | 63.0 (57.6–68.8) | 61.7 (56.3–67.6) | <0.001 |
| Sensory loss | 30.0 (26.2–34.0) | 36.9 (31.8–43.0) | 40.5 (34.6–47.3) | 28.8 (23.7–35.1) | 27.6 (22.4–34.1) | 24.8 (20.0–30.8) | 0.001 |
| Dysphagia | 37.3 (33.3–41.4) | 32.4 (28.1–37.3) | 29.6 (25.1–34.8) | 21.5 (17.1–26.9) | 19.2 (15.0–24.5) | 15.4 (11.8–20.1) | <0.001 |
| Dysphasia | 28.8 (25.2–32.8) | 28.7 (24.6–33.5) | 34.9 (29.8–40.7) | 31.8 (26.3–38.5) | 33.1 (27.3–40.1) | 25.7 (20.7–31.8) | 0.831 |
| Dysarthria | 42.5 (38.4–46.7) | 47.6 (42.4–53.6) | 50.3 (44.5–56.9) | 40.6 (34.8–47.4) | 45.6 (39.1–53.2) | 45.2 (38.7–52.7) | 0.893 |
| Incontinent | 42.7 (38.6–46.8) | 40.9 (36.5–45.9) | 39.2 (34.4–44.6) | 26.4 (21.8–31.8) | 28.1 (23.4–33.6) | 29.2 (24.4–34.9) | <0.001 |
| Total no. of impairments, median (IQR) | 3 (2–4) | 3 (2–5) | 3 (2–5) | 2 (1–3) | 2 (1–3) | 2 (1–3) | <0.001 |

[1]Chi-squared test for trend.

**Abbreviations:** AF, atrial fibrillation; MI, myocardial infarction; TIA, transient ischaemic attack; TOAST, Trial of Org 10172 in Acute Stroke Treatment

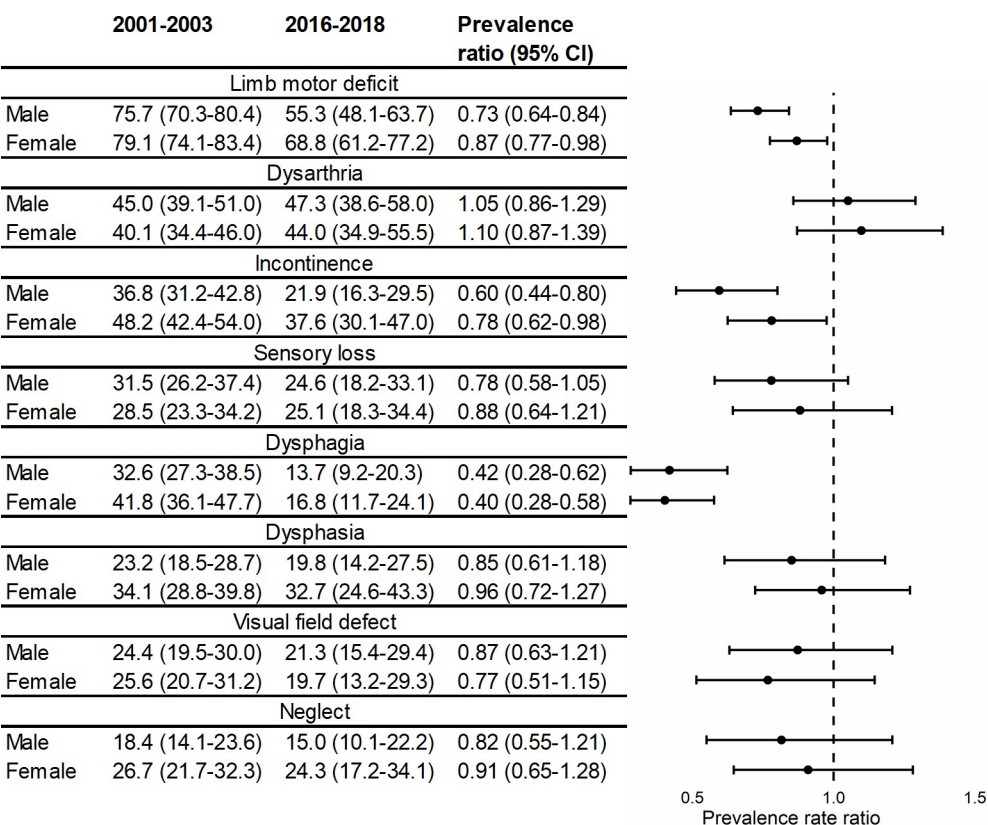

**Fig 2. Prevalence\* and prevalence ratios for acute stroke impairments over time, stratified by sex.** \*Adjusted for age, ethnicity, TOAST classification, and pre-stroke risk factors (hypertension, MI, AF, TIA, diabetes, high cholesterol, and smoking status) (*N* = 3,381). AF, atrial fibrillation; MI, myocardial infarction; TIA, transient ischaemic attack; TOAST, Trial of Org 10172 in Acute Stroke Treatment.

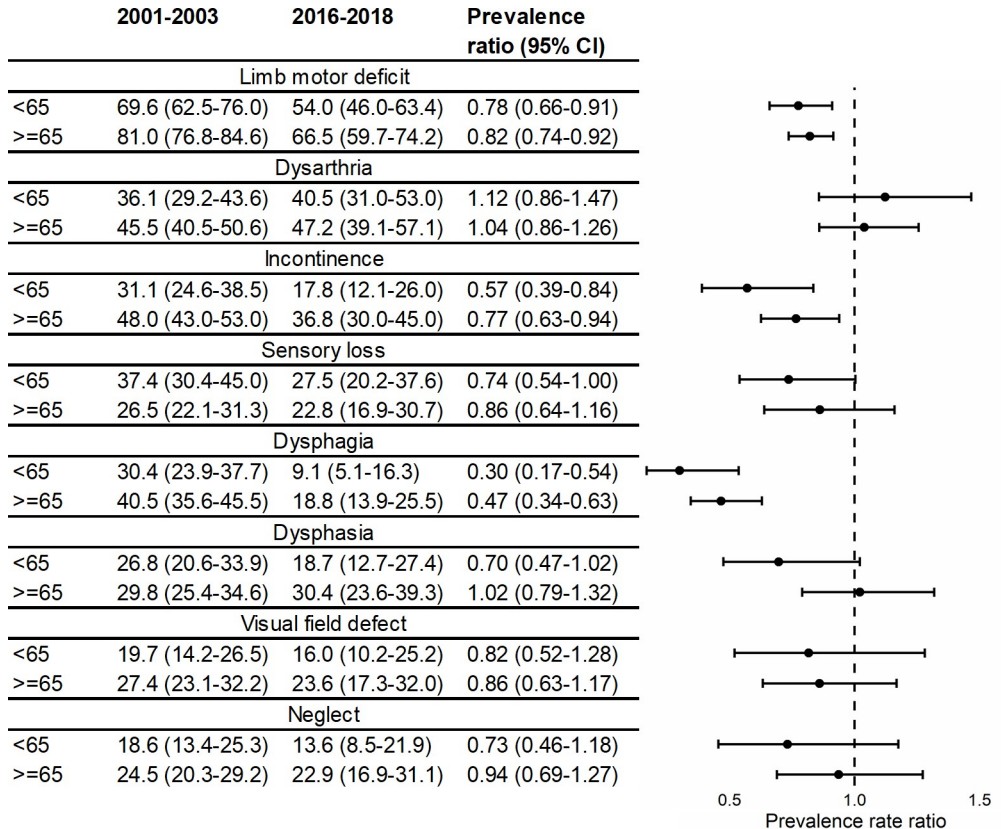

| | 2001–2003 | 2016–2018 | Prevalence ratio (95% CI) |
|---|---|---|---|
| | | Limb motor deficit | |
| <65 | 69.6 (62.5–76.0) | 54.0 (46.0–63.4) | 0.78 (0.66–0.91) |
| >=65 | 81.0 (76.8–84.6) | 66.5 (59.7–74.2) | 0.82 (0.74–0.92) |
| | | Dysarthria | |
| <65 | 36.1 (29.2–43.6) | 40.5 (31.0–53.0) | 1.12 (0.86–1.47) |
| >=65 | 45.5 (40.5–50.6) | 47.2 (39.1–57.1) | 1.04 (0.86–1.26) |
| | | Incontinence | |
| <65 | 31.1 (24.6–38.5) | 17.8 (12.1–26.0) | 0.57 (0.39–0.84) |
| >=65 | 48.0 (43.0–53.0) | 36.8 (30.0–45.0) | 0.77 (0.63–0.94) |
| | | Sensory loss | |
| <65 | 37.4 (30.4–45.0) | 27.5 (20.2–37.6) | 0.74 (0.54–1.00) |
| >=65 | 26.5 (22.1–31.3) | 22.8 (16.9–30.7) | 0.86 (0.64–1.16) |
| | | Dysphagia | |
| <65 | 30.4 (23.9–37.7) | 9.1 (5.1–16.3) | 0.30 (0.17–0.54) |
| >=65 | 40.5 (35.6–45.5) | 18.8 (13.9–25.5) | 0.47 (0.34–0.63) |
| | | Dysphasia | |
| <65 | 26.8 (20.6–33.9) | 18.7 (12.7–27.4) | 0.70 (0.47–1.02) |
| >=65 | 29.8 (25.4–34.6) | 30.4 (23.6–39.3) | 1.02 (0.79–1.32) |
| | | Visual field defect | |
| <65 | 19.7 (14.2–26.5) | 16.0 (10.2–25.2) | 0.82 (0.52–1.28) |
| >=65 | 27.4 (23.1–32.2) | 23.6 (17.3–32.0) | 0.86 (0.63–1.17) |
| | | Neglect | |
| <65 | 18.6 (13.4–25.3) | 13.6 (8.5–21.9) | 0.73 (0.46–1.18) |
| >=65 | 24.5 (20.3–29.2) | 22.9 (16.9–31.1) | 0.94 (0.69–1.27) |

**Fig 3. Prevalence* and prevalence ratios for acute stroke impairments over time, stratified by age.** *Adjusted for sex, ethnicity, TOAST classification, and pre-stroke risk factors (hypertension, MI, AF, TIA, diabetes, high cholesterol, and smoking status) (*N* = 3,381). AF, atrial fibrillation; MI, myocardial infarction; TIA, transient ischaemic attack; TOAST, Trial of Org 10172 in Acute Stroke Treatment.

implications for the long-term care of stroke survivors, highlighting a continuing need for specialist services to cater to the long-term disability associated with these acute impairments.

One reason for the reduction in acute impairments is a decrease in stroke severity, which we have shown with a reduction in the NIHSS score over the study period. Additional analysis has shown that this reduction in stroke severity is present in all stroke subtypes (S1 Table) and was not because of changes in the prevalence of any one aetiological subtype of stroke. We adjusted for stroke subtype when assessing the prevalence of acute impairments over time, suggesting that some other factor is driving the decrease in stroke severity.

Another plausible explanation for the reduction in acute impairments is improvements in primary preventive medicine over time. We have shown an increase in pre-stroke anticoagulation prescribing in people with AF. This is in line with reports from the Sentinel Stroke National Audit Programme (SSNAP) in England, showing that between 2013 and 2018, pre-stroke anticoagulation prescriptions for hospital stroke patients with AF increased from 38% to 61% nationally [22]. AF is a major risk factor for stroke that is associated with large cortical infarcts and increased stroke severity [23]. Anticoagulants, the recommended treatment for AF [24], have also been shown to reduce stroke severity [15]. Therefore, these improvements in anticoagulation prescribing may reduce the number of severe strokes associated with AF, resulting in this decline in acute impairments. During this same time period, however, we demonstrated declines in antiplatelet use. This may be explained by declines in the use of

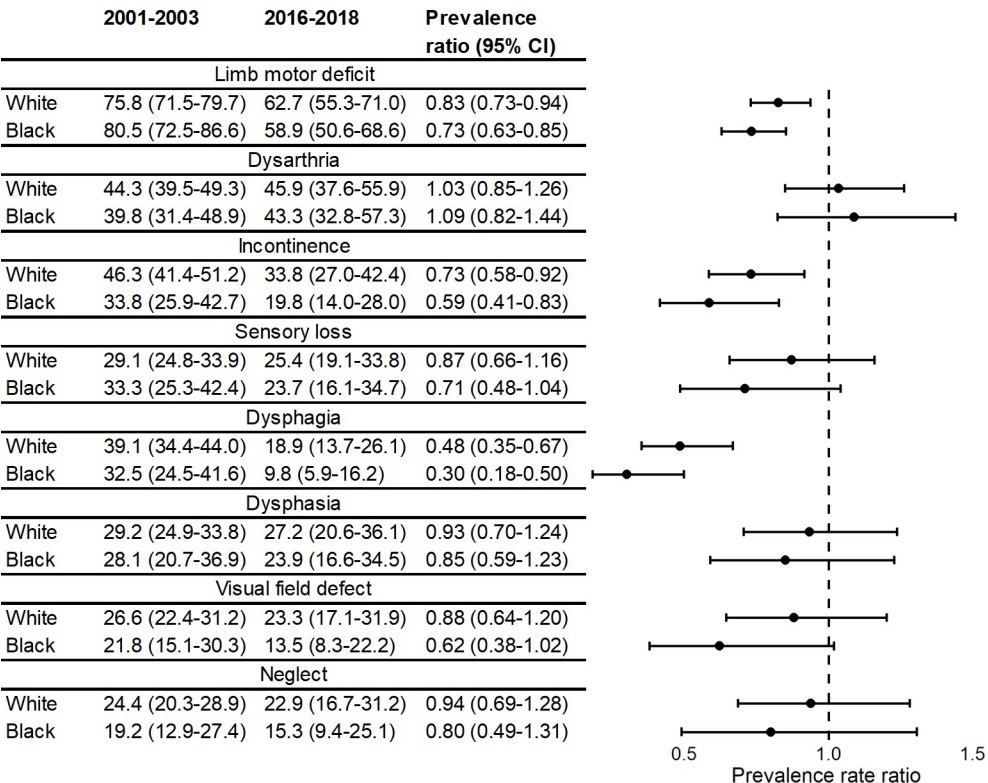

| | 2001-2003 | 2016-2018 | Prevalence ratio (95% CI) |
|---|---|---|---|
| Limb motor deficit | | | |
| White | 75.8 (71.5-79.7) | 62.7 (55.3-71.0) | 0.83 (0.73-0.94) |
| Black | 80.5 (72.5-86.6) | 58.9 (50.6-68.6) | 0.73 (0.63-0.85) |
| Dysarthria | | | |
| White | 44.3 (39.5-49.3) | 45.9 (37.6-55.9) | 1.03 (0.85-1.26) |
| Black | 39.8 (31.4-48.9) | 43.3 (32.8-57.3) | 1.09 (0.82-1.44) |
| Incontinence | | | |
| White | 46.3 (41.4-51.2) | 33.8 (27.0-42.4) | 0.73 (0.58-0.92) |
| Black | 33.8 (25.9-42.7) | 19.8 (14.0-28.0) | 0.59 (0.41-0.83) |
| Sensory loss | | | |
| White | 29.1 (24.8-33.9) | 25.4 (19.1-33.8) | 0.87 (0.66-1.16) |
| Black | 33.3 (25.3-42.4) | 23.7 (16.1-34.7) | 0.71 (0.48-1.04) |
| Dysphagia | | | |
| White | 39.1 (34.4-44.0) | 18.9 (13.7-26.1) | 0.48 (0.35-0.67) |
| Black | 32.5 (24.5-41.6) | 9.8 (5.9-16.2) | 0.30 (0.18-0.50) |
| Dysphasia | | | |
| White | 29.2 (24.9-33.8) | 27.2 (20.6-36.1) | 0.93 (0.70-1.24) |
| Black | 28.1 (20.7-36.9) | 23.9 (16.6-34.5) | 0.85 (0.59-1.23) |
| Visual field defect | | | |
| White | 26.6 (22.4-31.2) | 23.3 (17.1-31.9) | 0.88 (0.64-1.20) |
| Black | 21.8 (15.1-30.3) | 13.5 (8.3-22.2) | 0.62 (0.38-1.02) |
| Neglect | | | |
| White | 24.4 (20.3-28.9) | 22.9 (16.7-31.2) | 0.94 (0.69-1.28) |
| Black | 19.2 (12.9-27.4) | 15.3 (9.4-25.1) | 0.80 (0.49-1.31) |

**Fig 4. Prevalence* and prevalence ratios for acute stroke impairments over time, stratified by ethnicity.** *Adjusted for age, sex, TOAST classification, and pre-stroke risk factors (hypertension, MI, AF, TIA, diabetes, high cholesterol, and smoking status) (*N* = 3,381). AF, atrial fibrillation; MI, myocardial infarction; TIA, transient ischaemic attack; TOAST, Trial of Org 10172 in Acute Stroke Treatment.

antiplatelets in the AF population reported in England nationally [25], as aspirin, in particular, has been recognised as less effective for primary prevention of stroke than anticoagulants [26]. Therefore, the decrease in antiplatelet use may in part be because of a shift to increased use of anticoagulation that we have shown. Finally, we have also shown an increase in the prescription of statins pre-stroke, which have been shown to reduce stroke severity, and this improvement in primary prevention may also help explain the decline in impairments seen [16]. Previous research has also shown that prior antihypertensive use reduces stroke severity [17]; however, in this study we found that antihypertensive use has declined over time in those with hypertension. Therefore, this is unlikely to be associated with the decline in impairments we have shown.

Alternatively, the reductions in acute stroke impairments may be an artefact of changes in case ascertainment and differential changes in the recording of impairments over time. Better recognition of strokes over time may mean that more mild strokes are now being recognised and recorded, although we have shown that this has not affected the prevalence of any one particular stroke subtype.

For incontinence, we showed a decline from 43% in 2001–2003 to 29% in 2016–2018. Previous research showed that, between 1998 and 2002, there was no change in the prevalence of acute stroke incontinence, which remained at around 40% across the 5 years [27]. While this prevalence is comparable to what we found, the previous study found no decline in the impairment over the time period studied. Since this time point, we have shown a decline,

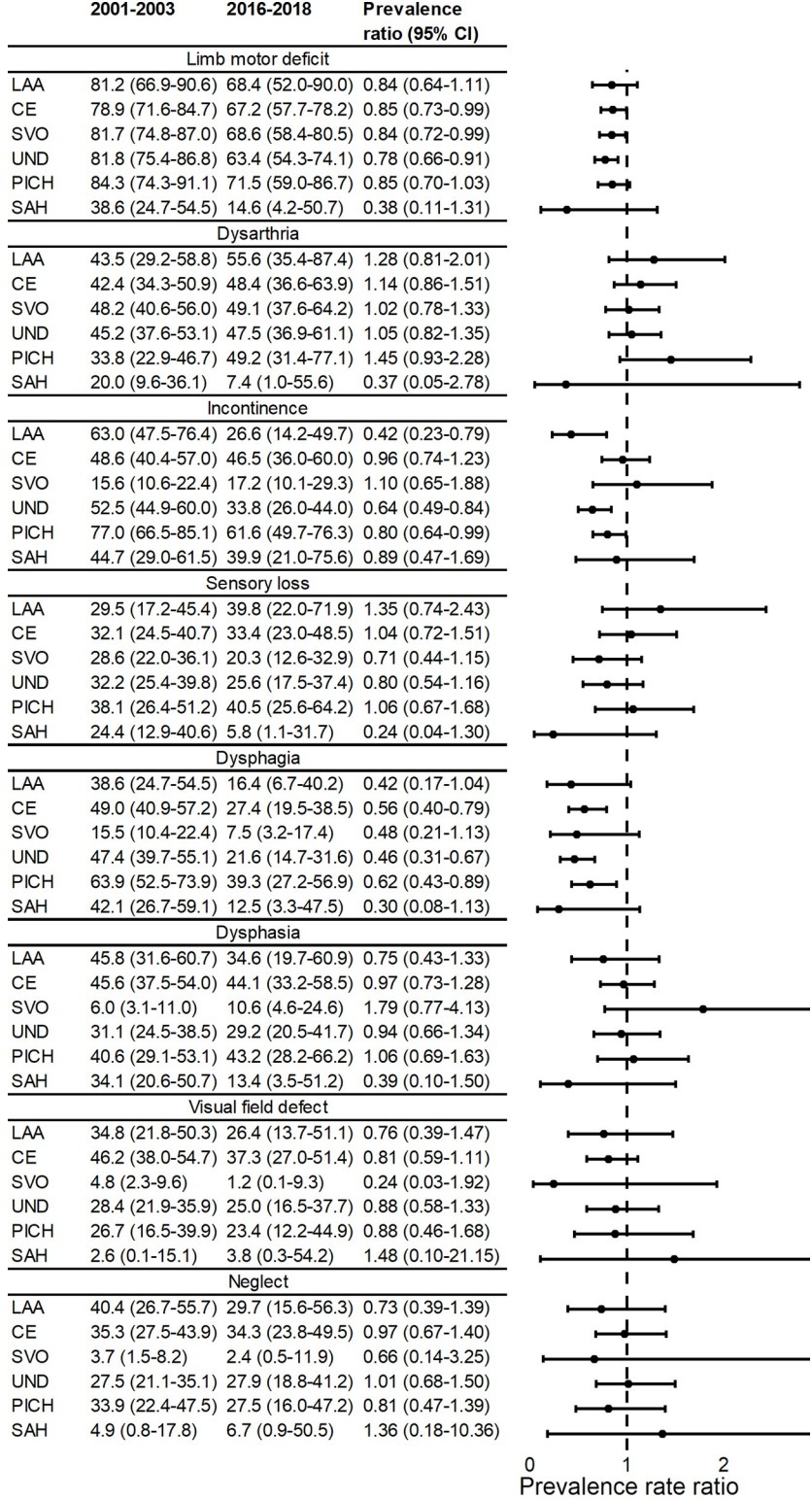

**Fig 5. Prevalence (adjusted for age, sex, and ethnicity) and prevalence ratios for acute stroke impairments over time, stratified by TOAST classification (*N* = 4,005).** CE, cardioembolism; LAA, large artery atherosclerosis; PICH, primary intracerebral haemorrhage; SAH, subarachnoid haemorrhage; SVO, small vessel occlusion; TOAST, Trial of Org 10172 in Acute Stroke Treatment; UND, undetermined aetiology.

**Table 3. Crude prevalence of acute impairments, stratified by TOAST classification (N = 3,922).**

| Acute impairment | Prevalence, N (%) | | | | | | p-Value[1] |
|---|---|---|---|---|---|---|---|
| | LAA (N = 371) | CE (N = 859) | SVO (N = 823) | UND (N = 1,151) | PICH (N = 525) | SAH (N = 193) | |
| **Limb motor deficit** | 272 (77.5) | 618 (76.1) | 614 (79.0) | 808 (74.5) | 375 (83.3) | 65 (42.2) | <0.001 |
| **Dysarthria** | 184 (53.2) | 388 (49.4) | 354 (45.5) | 504 (48.0) | 201 (48.0) | 24 (16.3) | <0.001 |
| **Incontinence** | 131 (36.5) | 394 (47.9) | 130 (16.4) | 439 (40.0) | 299 (61.8) | 88 (51.8) | <0.001 |
| **Sensory loss** | 133 (38.0) | 305 (39.0) | 221 (28.4) | 386 (36.3) | 195 (45.8) | 31 (20.7) | <0.001 |
| **Dysphagia** | 89 (27.4) | 318 (41.4) | 75 (10.6) | 303 (30.9) | 221 (53.0) | 45 (36.6) | <0.001 |
| **Dysphasia** | 141 (40.3) | 373 (46.7) | 67 (8.6) | 404 (37.9) | 207 (47.7) | 48 (32.0) | <0.001 |
| **Visual field defect** | 95 (27.2) | 308 (39.1) | 41 (5.3) | 293 (27.7) | 100 (23.8) | 6 (4.1) | <0.001 |
| **Neglect** | 117 (33.3) | 289 (36.9) | 45 (5.8) | 292 (27.7) | 134 (31.8) | 12 (8.0) | <0.001 |

[1]Chi-squared test for trend.

**Abbreviations:** CE, cardioembolism; LAA, large artery atherosclerosis; PICH, primary intracerebral haemorrhage; SAH, subarachnoid haemorrhage; SVO, small vessel occlusion; TOAST, Trial of Org 10172 in Acute Stroke Treatment; UND, undetermined aetiology

which is likely to have been revealed by studying an extended timeframe showing longer-term trends.

In the whole cohort, we showed an increase in Black stroke patients in South London, in line with previous research in the same population [28, 29]. However, comparing data from the 2001 and 2011 censuses and 2016 population estimates, the population of Black patients in the underlying population has not changed (24%, 25%, and 25%, respectively) [29, 30]. Therefore, this suggests an increase in stroke incidence in the Black population over time. Previous studies suggest that this may be due to unequal prevalence of risk factors in different ethnic groups [28, 31]. For example, we have shown an increase in the prevalence of diabetes and hypertension over time, which are known to affect the Black population more so than the White [28]. Research has also shown poorer management of hypertension in the Black population compared to the White, but any disparities in stroke-specific care are less clear [32, 33].

In our estimates of the prevalence of acute impairments, we adjusted for all demographic factors, therefore any changes in the stroke incidence by ethnicity was controlled for in the final prevalence estimates. In addition to this, we stratified by the demographic factors to explore whether the trends in the prevalence differed between subgroups. When stratifying by ethnicity, there was no difference in the trends in impairments between groups.

Stratifying by sex, we show that women have a higher prevalence of all stroke impairments compared to men, after adjusting for age. This is in line with previous research that women have more severe strokes than men [34]. In addition, women showed smaller declines in the prevalence of impairments—particularly limb function—over time, creating greater disparity between men and women. Evidence suggests that this is because women are at higher risk of hypertension and AF, and there is also evidence that women are less likely to be prescribed anticoagulants compared to men [35–37]. Given that preventive medicine may be a possible reason for improvements over time in acute impairments, this highlights the importance of ensuring that preventive medicine is as effective in women as it is in men to help narrow this gap in the future.

Stratifying by age, we found that for dysphasia, older people have made no improvements in the prevalence over time compared to younger people, after controlling for pre-stroke risk factors. This may indicate that improvements over time in preventive medicine are disproportionately made in the younger population. Evidence has also shown that older people are given poorer post-stroke care [38]. This further widens the gap between old and young in long-term

outcomes. Dysphasia, in particular, reduces the ability to partake in social activities, affecting long-term quality of life, which is particularly important for the older population [39, 40].

Finally, we provide novel data highlighting how the burden of acute stroke impairments differs between stroke subtype. We have found that strokes of undetermined aetiology have high levels of impairment comparable to that of LAA and CEs. This group is also the most common, at around 30% at each time period, comparable to levels identified in previous research [41]. Strokes of undetermined aetiology are most commonly due to covert AF, highlighting the need for better detection and prevention in this subgroup [42]. We also found that the dominant impairment in SVOs was limb motor deficit, with all other impairments at substantially lower levels. Given that long-term outcome is better in SVOs compared to LAA, CEs, and undefined strokes [43], this highlights that all other impairments are significant in defining the prognosis of these patients.

## Limitations

A limitation of this study is that some impairments were collected using methods separate from others. Dysphagia and incontinence were collected using their own assessments, whereas all other impairments were measured using the multicomponent NIHSS. While the methods to assess these impairments have not changed over time, there may have been differential changes in the timing and speed at which assessments were undertaken or recorded. For example, in patients eligible for thrombolysis, impairments may not have been recorded until after acute stroke care, resulting in an apparent improvement in impairments.

Another limitation of this study is that we only looked at the physical impairments caused by the stroke and not any cognitive and psychological impairments. This is because cognitive data are collected days after the stroke, rather than in the acute phase. However, it is well known that these are a substantial burden faced by stroke patients, which has previously been studied in the SLSR [44]. We also lack data on the psychological significance of these disabilities for patients and how each physical impairment impacts the quality of life of stroke survivors differently, which will also be important in the care and planning of support for stroke survivors and warrants further research.

Finally, this study was conducted on a population-based cohort based in inner-city London. While this is a strength, the demographics of our cohort differ from that of the general UK population, and therefore the overall findings may not be directly comparable to other cohorts. However, we have provided results for subgroups of our population, which allows for interpretation by other populations with different demographic makeups.

## Conclusion

Our study highlights to policymakers, care providers, and patients the plethora of complex impairments that patients and their care providers must cope with. We provide novel evidence of a large, longitudinal study for a decline in the prevalence of limb motor deficit, incontinence, and dysphagia over an 18-year period, which may be driven by improvements in preventive medicine. Declines in limb motor deficit were greater in men than in women, and improvements were also greater in younger compared to older patients.

Acute impairments are the strongest predictors of long-term outcome, and we have highlighted the potential impact of anticoagulation on reducing these impairments. Therefore, we shed light on subgroups of patients for whom preventive medicine, post-stroke care, and specialist services are a priority. Further research should identify how comorbidities interact with acute stroke impairments in all patient subgroups to ensure that appropriate care is provided to optimise long-term quality of life.

## Supporting information

**S1 STROBE Checklist. Checklist of items that should be included in reports of cohort studies.** STROBE, Strengthening The Reporting of OBservational studies in Epidemiology.
(DOCX)

**S1 Table. Trends in the median NIHSS score over time, stratified by aetiological subtype of stroke.** NIHSS, National Institutes of Health Stroke Scale.
(DOCX)

**S1 Fig. Trends in the prevalence of acute stroke impairments, crude and adjusted for age, sex, ethnicity, TOAST classification, and pre-stroke risk factors (hypertension, MI, AF, transient ischemic attack, diabetes, high cholesterol, and smoking status).** AF, atrial fibrillation; MI, myocardial infarction; TOAST, Trial of Org 10172 in Acute Stroke Treatment.
(TIF)

**S2 Fig. Trends in the prevalence of acute impairments over time, stratified by sex, age, and ethnicity, adjusted for TOAST classification, pre-stroke risk factors (hypertension, MI, AF, TIA, diabetes, high cholesterol, and smoking status), and age, sex, and ethnicity where applicable.** AF, atrial fibrillation; MI, myocardial infarction; TIA, transient ischaemic attack; TOAST, Trial of Org 10172 in Acute Stroke Treatment.
(TIF)

**S3 Fig. Trends in the prevalence of acute impairments, stratified by TOAST classification, adjusted for age, sex, and ethnicity.** TOAST, Trial of Org 10172 in Acute Stroke Treatment.
(TIF)

## Author Contributions

**Conceptualization:** Ajay Bhalla, Anthony G. Rudd, Charles D. A. Wolfe, Yanzhong Wang.

**Formal analysis:** Amanda Clery.

**Funding acquisition:** Charles D. A. Wolfe, Yanzhong Wang.

**Supervision:** Ajay Bhalla, Yanzhong Wang.

**Writing – original draft:** Amanda Clery.

**Writing – review & editing:** Ajay Bhalla, Anthony G. Rudd, Charles D. A. Wolfe, Yanzhong Wang.

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
