## [Decision Letter · Decision Letter 0]

2 Jul 2020

Dear Dr. Clery,

Thank you very much for submitting your manuscript "The trends in prevalence of acute stroke impairments: Analysis using the South London Stroke Register" (PMEDICINE-D-20-02083) for consideration at PLOS Medicine. 

[LINK]

In light of these reviews, I am afraid that we will not be able to accept the manuscript for publication in the journal in its current form, but we would like to consider a revised version that addresses the reviewers' and editors' comments. Obviously we cannot make any decision about publication until we have seen the revised manuscript and your response, and we plan to seek re-review by one or more of the reviewers. 

We expect to receive your revised manuscript by Jul 23 2020 11:59PM. Please email us (plosmedicine@plos.org) if you have any questions or concerns.

We look forward to receiving your revised manuscript. 

Sincerely,

Emma Veitch, PhD

PLOS Medicine

On behalf of Clare Stone, PhD, Acting Chief Editor,

PLOS Medicine

plosmedicine.org

*In the Abstract Methods and Findings section, please include a brief note summarising any key limitations of the study methods.

*At this stage, we ask that you include a short, non-technical Author Summary of your research to make findings accessible to a wide audience that includes both scientists and non-scientists. The Author Summary should immediately follow the Abstract in your revised manuscript. This text is subject to editorial change and should be distinct from the scientific abstract. Please see our author guidelines for more information: https://journals.plos.org/plosmedicine/s/revising-your-manuscript#loc-author-summary

*Please clarify in the paper if the analytical approach followed here corresponded to one laid out in a prospective study protocol or analysis plan? Please state this (either way) early in the Methods section.

*We'd suggest using an appropriate reporting guideline (eg STROBE - https://www.equator-network.org/reporting-guidelines/strobe/) - to guide reporting of study methods and findings; if doing this please ensure you upload the completed STROBE checklist as supporting information alongside the revised paper.

*At the moment the Discussion section doesn't explicitly include a clear Strengths and Limitations section; with respect to the latter this should include a summary of any specific biases inherent in the study methods which might have prevented the observed effects from closely approximating the "real" effects, and if so in what direction those biases might have have affected observed outcomes (eg towards null or away from null).

Comments from the reviewers:

Reviewer #1: I confine my remarks to statistical aspects of this paper. These were well done and I recommend publication

Peter Flom

Reviewer #2: Clery et al. report the trends of acute stroke impairments overtime using population-based data. Limb motor deficits, dysphagia and incontinence declined between 2001 and 2018. Interestingly, the prevalence of all pre-stroke risk factors, except for vascular disease, increased over time. The authors hypothesize that the increased use of anticoagulants and statins could count, at least in part, for the decrease of acute impairments.

- This study analyzes ischemic stroke, ICH and SAH. These diseases are different in etiology, severity and prognosis. The article is mostly focused on ischemic stroke. The authors may want to consider concentrating in ischemic stroke solely.

- Can the authors provide data on gaze paresis and dysarthria?

- Why did the authors exclude the data from 1995 to 2001?

- The decrease in the use of anti platelets may be related to new data showing that there is no benefit on its use for primary prevention 

- The results are important for policy makers, but I believe these data are population specific and cannot be extrapolated. also, the decrease in stroke severity, and its association with the use of antithrombotics has already been shown.

- "We have found that strokes of undetermined aetiology have high levels of impairments comparable to that of LAAs and CEs." Stroke of undetermined source (aka ESUS) are mostly due to covert afib or carotid plaques < 50%. This may also explain the reported findings.

Reviewer #3: This paper describes the trends in prevalence of acute stroke impairments in a large stroke registry. The results are interesting, and the analysis is convincing. I have just a few minor questions and comments to Authors.

1) page 6, line 83. Please give values in mg, most of us readers do not use mmols.

2) page 7 line 88. Despite relevant negative comments about (see for instance Stroke 36(4):902-4 · May 2005 "Time to burn Toast"), Toast classification is still widely used. However, I suggest Authors, if at all possible, to use OCSP classification, and also to look at Esus cases, which could have different level of impairment. 

3) page 7 line 92. As for haemorrhages, could Authors differentiate between typical and atypical forms? It seems that the latter are increased in recent years, and the impact on impairment may well be different.

4) page 8 line 112. Authors did not adjust for smoking and cholesterol levels. Is there a specific reason for that? If not, could they add these two factors to their analysis?

5) Results line 134. Mean age looks quite low, as compared to the one found in different European studies. Do Authors think that this fact could limit the external validity of their results?

6) Results line 152. The decrease in severity is evident. Hower, could it be due to more mild cases being admitted, or to less severe cases being admitted? is any other facility available for stroke patient in the study area? Have Authors any information on the number of patients staying at home with their stroke?

[LINK]

---

## [Decision Letter · Decision Letter 1]

12 Aug 2020

Dear Dr. Clery,

Thank you very much for re-submitting your manuscript "The trends in prevalence of acute stroke impairments: A population-based cohort study using the South London Stroke Register" (PMEDICINE-D-20-02083R1) for review by PLOS Medicine.

I have discussed the paper with my colleagues and the academic editor and it was also seen again by reviewers. I am pleased to say that provided the remaining editorial and production issues are dealt with we are planning to accept the paper for publication in the journal.

[LINK]

We look forward to receiving the revised manuscript by Aug 19 2020 11:59PM. 

Sincerely,

Clare Stone, PhD

Managing Editor 

PLOS Medicine

plosmedicine.org

Requests from Editors:

Competing interests: Please mention the YW is a paid statistical referee for PLOS Medicine, but had no role in the peer review of this paper. 

Title – I suggest removing ‘The’

Abstract – Please include summary demographic information and provide p values for 95% Cis (here and elsewhere). Also, Please quote one further limitation in the abstract

Author Summary – briefly please say what the 5 conditions are that were unaffected.

Please include a space before the opening square brackets for refs. 

STROBE – please submit supp files individually, including this one and also sections and paragraphs should be used instead of pages as these change during formatting and revisions. 

- at line 35, suggest "In this study, we found that stroke patients in the SLSR had a complexity ..." (also similar amendments at line 56)

- Poor punctuation at line 312; in fact, this whole paragraph on ethnic disparities is quite hard to understand. Do you mean that the proportion of Black people in the relevant population has not changed, and therefore that a higher proportion of Black people than White have risk factors and suffer strokes in South London? Please clarify. 

Comments from Reviewers:

Reviewer #2: My concerns have been appropriately addressed.

Thank you for the opportunity of reviewing your paper.

Reviewer #3: Authors were kind enough to consider all my points, and, as far as possible, to follow my suggestions. I have no further question or comment.

[LINK]

---

## [Editor Report · Decision Letter 2]

31 Aug 2020

Dear Dr Clery, 

On behalf of my colleagues and the academic editor, Dr. Joshua Willey, I am delighted to inform you that your manuscript entitled "Trends in prevalence of acute stroke impairments: A population-based cohort study using the South London Stroke Register" (PMEDICINE-D-20-02083R2) has been accepted for publication in PLOS Medicine. 

PRODUCTION PROCESS

PRESS

PROFILE INFORMATION

Thank you again for submitting the manuscript to PLOS Medicine. We look forward to publishing it. 

Best wishes, 

Clare Stone, PhD

Managing Editor 

PLOS Medicine

plosmedicine.org